# An Outbred Guinea Pig Disease Model for Lassa Fever Using a Host-Adapted Clade III Nigerian Lassa Virus

**DOI:** 10.3390/v15030769

**Published:** 2023-03-17

**Authors:** Yvon Deschambault, Geoff Soule, Levi Klassen, Angela Sloan, Jonathan Audet, Kim Azaransky, Abdulmajid S. Musa, Adama Ahmad, Afolabi M. Akinpelu, Nwando Mba, Derek R. Stein, Marc Ranson, Muhamad Almiski, Kevin Tierney, Gabor Fischer, Mable Chan, David Safronetz

**Affiliations:** 1Special Pathogens, National Microbiology Laboratory Branch, Public Health Agency of Canada, Winnipeg, MB R3E 3M4, Canada; 2Nigerian Centre for Disease Control, Jabi, Abuja 900108, Nigeria; 3Department of Pathology, University of Manitoba, Winnipeg, MB R3E 3P5, Canada; 4Shared Health Diagnostic Services, Winnipeg, MB R3C 3H8, Canada; 5Department of Medical Microbiology and Infectious Diseases, University of Manitoba, Winnipeg, MB R3E 0J9, Canada

**Keywords:** Lassa virus, disease modelling, medical countermeasures, infectious diseases

## Abstract

Nigeria experiences annual outbreaks of Lassa fever (LF) with high case numbers. At least three clades of Lassa virus (LASV) have been documented in Nigeria, though recent outbreaks are most often associated with clade II or clade III viruses. Using a recently isolated clade III LASV from a case of LF in Nigeria in 2018, we developed and characterized a guinea pig adapted virus capable of causing lethal disease in commercially available Hartley guinea pigs. Uniform lethality was observed after four passages of the virus and was associated with only two dominant genomic changes. The adapted virus was highly virulent with a median lethal dose of 10 median tissue culture infectious doses. Disease was characterized by several hallmarks of LF in similar models including high fever, thrombocytopenia, coagulation disorders, and increased inflammatory immune mediators. High viral loads were noted in all solid organ specimens analyzed. Histological abnormalities were most striking in the lungs and livers of terminal animals and included interstitial inflammation, edema, and steatosis. Overall, this model represents a convenient small animal model for a clade III Nigeria LASV with which evaluation of specific prophylactic vaccines and medical countermeasures can be conducted.

## 1. Introduction

Lassa mammarenavirus (LASV) is a zoonotic pathogen that causes a wide spectrum of symptoms in humans, from seemingly asymptomatic or non-descript illness to severe and life-threatening disease with multi-organ failure, referred to as Lassa fever (LF) [1,2,3]. In nature, LASV is maintained primarily in *Mastomys natalensis*, though other small mammals may also serve in that capacity [4,5]. LASV is endemic in several countries in West Africa with cases of LF detected annually in Sierra Leone, Liberia, Guinea, and Nigeria [6]. The annual incidence of LF and particularly LASV infection remains unclear but is commonly estimated at 300,000–500,000 cases per annum based on early clinical studies [4,7]. Likewise, the case fatality rate for LF varies widely from an estimated 1–2% of all LASV infections, to between 30–50% in laboratory confirmed, and hospitalized cases [3,8,9,10,11]. Over the last few years, increased numbers of LF cases have been recognized, particularly in Nigeria, which has documented annual outbreaks with mortality rates exceeding historical observations [12,13,14,15]. Currently, there are no approved prophylactic vaccines to prevent LF and therapeutic options are limited to ribavirin, which has disputed efficacy in improving patient outcomes [16]. For this reason, LASV is considered a priority pathogen for the rapid development of effective countermeasures to treat and prevent disease in humans [17].

Animal models of disease provide an important means for the pre-clinical development of medical countermeasures against infectious diseases, particularly those that are rare, emerging, or of high lethality. For many etiological agents of viral hemorrhagic fevers (VHFs), inoculation of non-human primates (NHPs) with authentic, cell culture isolated, viruses result in severe diseases, which recapitulates the anticipated human disease. However, development of small animal models for the same agents often relies on the use of specialized immunocompromised or specifically inbred animal species. One approach that has been reliably used to circumvent the requirement for these specialized species is serial passaging viruses in commercially available outbred species, creating rodent-adapted variants. Modelling LF and LASV infections in small mammals is most commonly achieved in inbred strain 13 guinea pigs [18]. However, procurement of these animals can be difficult and often relies on maintenance of in-house breeding colonies to support research studies. Previously, a guinea pig adapted LASV strain Josiah was described in Hartley guinea pigs and utilized for the evaluation of vaccines against LASV infection [19]. To expand small animal models for contemporary strains of LASV with outbreak potentials, we sought to adapt a recently isolated clade III LASV strain obtained from a clinical case in Nigeria during the 2018 LF epidemic.

## 2. Methods

### 2.1. Ethics and Biosafety

Animal studies were approved by the Animal Care Committee of the Canadian Science Centre for Human and Animal Health (CSCHAH). All procedures were performed by trained personnel in a Canadian Council on Animal Care (CCAC) approved facility according to CCAC guidelines. Work involving LASV was performed in a Biosafety Level 4 (BSL4) laboratory of the Public Health Agency of Canada. When required, materials were inactivated according to approved procedures for subsequent analyses.

### 2.2. Animal

Adult outbred female Hartley guinea pigs (*Cavia porcellus*) weighing approximately 500 g were purchased from Charles River. Adult, inbred strain 13 guinea pigs weighing approximately 800 g were sourced from an in-house breeding colony maintained at the University of Manitoba. Prior to challenge, animals were subcutaneously implanted with a BMDS IPTT-300 programmable temperature transponder (BMDS, Seaford, DE, USA) into the intra-scapular region. All experimental manipulations were conducted on sedated animals (3–5% inhalational isoflurane (WDDC, Edmonton, AB, CD) maintained in medical oxygen). Animals were euthanized via exsanguination via cardiac puncture while under deep anesthesia followed by a thoracotomy.

### 2.3. LASV Isolate

The isolation and description of the clade III LASV isolate utilized in these studies, NML-61, has previously been published [20]. Briefly, the virus was isolated from a non-lethal, symptomatic case of LF originating from Plateau state, Nigeria, during the 2018 outbreak.

### 2.4. In Vivo Challenge and Passaging

As an initial assessment of disease progression and lethality, groups of six inbred strain 13 and 8 outbred Hartley guinea pigs were infected with 1 × 10^5^ TCID_50_ units of the Vero-p1 isolate of LASV NML-61 via intraperitoneal (i.p.) injection. Post-infection, animals were observed daily for clinical signs of infection including ruffled fur, labored breathing, weight loss, increased body temperature, and recumbence. Animals were humanely euthanized if they reached a pre-determined endpoint according to an approved clinical scoring sheet.

To achieve uniform disease progression in the commercially available Hartley guinea pigs, samples were collected from animals that demonstrated advanced signs of disease requiring euthanasia for passaging. Briefly, tissues (lung, liver, spleen, and kidney) from a single animal were collected and individually mechanically homogenized (10% *w/v*) in serum free media using a 5 mm stainless steel bead and the Omni-Inc Bead Rupture Elite homogenizer (4 m/s for 30 s). Homogenized tissues were clarified (1500× *g* for 10 min) three times and the supernatants pooled, diluted 1/10 and used as inoculum for the next passage. Each passage consisted of four Hartley guinea pigs infected via i.p. injection of the previous passages tissue homogenate. Post-inoculation animals were observed daily for signs of disease progression as outlined above. Passaging continued until two consecutive passages achieved 100% lethality in Hartley guinea pigs, after which the pooled tissue homogenates from a single animal was used to isolate and titer (see methods below) the adapted virus (referred to as NML-61/GPA) using standard Vero E6 cell culture techniques. Following isolation of the adapted virus, two groups of six Hartley guinea pigs were inoculated with either NML-61/GPA or the tissue homogenates from which it was isolated to ensure the single cell culture passage did not alter disease progression of the isolate.

### 2.5. Determination of 50% Lethal Dose (LD_50_)

Six groups of six Hartley guinea pigs were infected with 10-fold serial dilutions (10^1^ to 10^6^ TCID_50_) of the NML-61/GPA via i.p. injection. Animals were monitored daily for signs of disease progression and euthanized once they reached a predetermined humane endpoint.

### 2.6. Temporal Infection Kinetics of NML-61/GPA

Twenty-four Hartley guinea pigs were infected with 100x the LD_50_. On days 1, 3, 6, 9, 12, and 15 post-infection, four animals were randomly selected for analysis. Tissues (lung, liver, spleen, and kidney) were collected for viral titrations and histopathological analysis. Blood and serum were collected for complete blood count, biochemistry analysis, and coagulation assays.

### 2.7. Viral Titrations (TCID_50_)

Infectious viral titers were determined in tissue specimens (lungs, liver, spleen, kidney, and blood) collected during necropsy using standard tissue culture infectivity assays. Briefly, for solid organs, a 30 mg piece of tissue was mechanically homogenized and clarified as outlined above and used to prepare 10-fold serial dilutions in sterile DMEM (Gibco/Fisher Scientific, Ottawa, ON, CD) supplemented with 1% FBS (Corning, Woodland, CA, USA), 1% Penicillin/Streptomycin, and 1% L-glutamine (all from Gibco). Serial dilutions of blood samples were prepared in the same manner. One-hundred microliters of each dilution were added to triplicate wells of 80% confluent monolayers of Vero E6 cells seeded the night prior in 96 well plates. Cells were incubated for 5–7 days at 37 °C with 5% CO_2_. Cytopathic effect (CPE) was assessed on consecutive days and the 50% tissue culture infectious dose (TCID_50_) calculated by the Spearman–Karber method. Due to low volumes, whole blood samples were extracted for RNA and qualitative molecular detection of LASV was conducted using an in-house, validated diagnostic RT-PCR test with primers LaV-F (5′- CCACCATYTTRTGCATRTGCCA) and LaV-R (5′-GCACATGTNTCHTAYAGYATGGAYCA), and probe LaV-P (5′-FAM-AARTGGGGYCCDATGATGTGYCCWTT).

### 2.8. Hematology, Biochemistry and Coagulation

Hematology was conducted on EDTA-treated cardiac blood using a Vetscan HM5 (Abaxis, Parsippany, New Jersey, USA). Serum biochemistries were monitored using comprehensive diagnostic profile cassettes on a Vetscan VS2 Analyzer (Abaxis). Citrate-treated plasma was utilized to assess fibrinogen, aPTT, PT, and Thrombin on a Satellite Max instrument (Stago, Mississauga, ON, Canada).

### 2.9. Cytokine Responses

Serum concentrations of nine cytokines (IFN gamma, INF alpha, TNF alpha, TGF beta 1, IL-2, IL-4, IL-6, IL-8, and IL-1 beta) were monitored in samples collected during the temporal kinetics study. Samples were gamma irradiated (5 MRad from a Cobalt-60 source) and run in duplicate on specific enzyme-linked immunosorbent assays according to the manufacturer’s specifications (Abclonal, Woburn, MA, USA).

### 2.10. Histopathology

Immediately post collection, tissue specimens were submerged in 10% neutral buffered formalin and fixed for a minimum of 28 days. Tissues were placed in cassettes and processed with a Sakura VIP-6 Tissue Tek on a 12-hr automated schedule, using a graded series of ethanol, xylene, and PureAffin. Embedded tissues were sectioned at 5 µm and dried overnight at 42 °C prior to staining with hematoxylin and eosin (H&E) according to standard histopathological methods. Slides were blindly evaluated by a clinical pathologist.

### 2.11. Illumina Sequencing

The genomic sequence of NML-61/GPA was determined on a MiniSeq instrument as previously described [20]. The resultant sequence was compared to that from the original viral inoculum to determine potential mutations responsible for lethal adaption. The data was analyzed by trimming adapter and low-quality sequences using Trim Galore. The reads were aligned to the guinea pig genome and non-aligned reads were extracted. These reads were aligned to the reference sequence for LASV NML-61. Alignment and indel qualities were added using LoFreq3, which was also used to call the mutations. A minimum sequence depth of 30 reads and minimum frequency of 5% was required for a mutation to be reported.

## 3. Results

### 3.1. Generation of a Guinea Pig Adapted Clade III LASV

In an initial assessment of virulence, a first passage (p1) clinical isolate (NML-61) propagated in Vero E6 cells was used to infect six inbred strain 13 and eight outbred guinea pigs. The isolate was obtained from a non-lethal clinical case of LF diagnosed in Nigeria in 2018 and was genetically confirmed as a clade III LASV [20]. Infection of strain 13 guinea pigs was uniformly lethal within 16 days post-infection. Moribund strain 13 guinea pigs began losing weight by day 6–7 post-infection and had elevated temperatures by day 9 post-infection (Figure 1). In Hartley guinea pigs, trends in body temperature mirrored those observed in strain 13 guinea pigs though body weights remained relatively static until after day 5 when slight increases were observed. Nevertheless, the clinical clade III LASV isolate was 50% lethal in Hartley guinea pigs (Figure 1).

NML-61/GPA was generated by serially passaging the virus four times in Hartley guinea pigs. Uniform lethality was observed in the final two passages, after which the virus was isolated from tissue homogenates from a single terminally ill animal on Vero E6 cells. In the final assessment, groups of six Hartley guinea pigs were infected with NML-61/GPA or the tissue homogenates from which it was derived. Both viral preparations were uniformly lethal within 12–13 days of inoculation. The mean time to death was calculated at 12 days in animals inoculated with tissue homogenates of the final passage, versus 13.5 days in animals inoculated with the cell culture derived NML-61/GPA (Figure 2).

### 3.2. Determination of the 50% Lethal Dose

Following confirmation that the Vero isolated NML-61/GPA retained uniform lethality in Hartley guinea pigs, the median (50%) lethal dose (LD_50_) was calculated. Groups of 6 animals were infected with 10-fold serial dilutions of NML-61/GPA ranging from 1 × 10^6^ to 1 × 10^1^ TCID_50_. Severe disease requiring humane euthanasia was noted in guinea pigs between 9- and 17-days post-infection, largely in a dose dependent trend. The LD_50_ was calculated to be 10^1^ TCID_50_ (Figure 2).

### 3.3. Full Genome Sequence Analysis

RNA was extracted from the supernatant of the Vero-isolated NML-61/GPA and the full genomic sequence was determined by Illumina sequencing. Comparison to the sequence generated from the original (non-guinea pig adapted) clinical isolate of NML-61 propagated one time on Vero E6 cells (MZ169798 (L), MZ169799 (S)) revealed two mutations at a frequency of greater than 50%. One mutation occurred at position 228 of the nucleoprotein (isoleucine to valine mutation) and one at position 1221 in the polymerase (asparagine to aspartic acid mutation). Several other lower frequency mutations were also observed and are summarized in Table 1.

### 3.4. Clinical Observations in Guinea Pigs Infected with NML-61/GPA

The first indication of ensuing disease in infected guinea pigs was elevated body temperatures exceeding 40 °C around day 9 post-infection. Temperatures remained elevated and often exceeded 41 °C until immediately preceding death, at which point a precipitous drop was noted. Weight loss generally occurred in parallel with elevated temperatures with most animals losing between 10 and 20% of their maximum experimental weight. Overt physical signs of infection were minimal in this model and were largely limited to reduced activity and increase respiration rates occurring in the terminal stages of disease. At no point were neurological signs apparent in infected animals.

### 3.5. Viral Titers in Solid Organs

Infectious LASV was detected in lung, kidney, and spleen specimens from two or three animals as early as day 3 post-infection (Figure 3). By day 6 post-infection, all organ specimens evaluated from each infected animal had detectable infectious LASV with increasing titers observed until the terminal time-point in lung, liver, and kidney samples. In contrast, LASV titers in spleen specimens remained relatively static on days 6 and 9 post-infection and decreased in the days immediately preceding death (Figure 3). Consistent with these observations, RT-PCR positive blood samples were first noted on day 3 post-infection (3 of 4 animals), with uniform positivity detected on and after day 6 post-infection. Although the molecular assay was not quantitative, the decreasing Ct values noted after day 6 imply increasing viral titers up until the terminal time point (Appendix A).

### 3.6. Hematology, Serum Biochemistries, and Coagulation Parameters

Hematological analysis of white blood cells (lymphocytes, monocytes, neutrophils) as well as red blood cells (RBC) and hemoglobin (HGB) revealed no overt pattern of change during the course of the study (Appendix A). Significant decreases in the red cell distribution width (RDW) were noted on days 9 and 12 post-infection as well as mean corpuscular volume (MCV) and mean corpuscular hemoglobin (MCH), though only at the day 12 post-infection time point (Figure 4). Mean corpuscular hemoglobin concentration (MCHC) remained unchanged throughout the course of the study. The most striking hematological abnormality noted was a decline in platelet (PLT) counts on days 6, 9, and 12 post-infection which corresponded with increases in values for mean platelet volume (MPV) and platelet distribution width (PDW) on days 6, 9, and 12, and days 9 and 12 post-infection, respectively (Figure 4).

Notable changes in serum biochemistries included significant decreases in alkaline phosphatase (ALP) at day 12 post-infection, as well as phosphatase (PHOS) at both day 9 and 12 time-points. A transient increase in blood urea nitrogen (BUN) was observed at day 3 post-infection and increased levels of globulin (GLOB) at days 9 and 12 post-infection were also observed (Figure 5). The remaining parameters, including alanine aminotransferase (ALT), amylase (AMY), total bilirubin (TBIL), calcium (CA), creatinine (CRE), glucose (GLU), sodium (NA), potassium (K), and total protein (TP), remained relatively unchanged during infection (Figure 5, Appendix A).

Coagulation profiles demonstrated general increases in prothrombin time (PT), activated partial thromboplastin time (aPTT) and Fibrinogen up until day 9 post-infection, with decreases in values observed after onset of severe disease (Figure 6). Thrombin times (TT) remained relatively unchanged throughout the course of the study.

### 3.7. Host Immune Responses

Nine cytokines were monitored in serum samples collected at regular intervals post-infection using guinea pig specific ELISA methodologies (Figure 7). Concentrations of IFN alpha and IL-8 increased gradually over the course of infection, whereas levels of IFN gamma, IL-2 and TNF alpha were relatively stable until day 9 post-infection at which point dramatic increases were observed. IL-4, IL-6, and IL-1 beta peaked by day 9 post-infection then dropped immediately prior to severe disease onset. Similarly, TGF-beta levels were relatively static until after day 6 post-infection at which point concentrations decreased.

### 3.8. Pathology and Histopathological Analysis

A comprehensive histologic examination was performed on sections obtained from four organs (liver, lung, kidney, and spleen) from each animal. The morphologic features in the liver and lung showed substantial differences and progression in the different animal groups, which correlated with the time of the euthanasia. The most striking findings were the fatty changes in the liver and the interstitial inflammation in the lung, accompanied with the dramatic decrease of open alveolar surfaces in some animals.

Liver specimens collected at days 1 and 3 post-infection were largely unremarkable. Early histopathological abnormalities consisted of minimal necrosis with occasional hepatocellular dropout noted in animals euthanized at day 6. Minimal necrosis remained present in animals sampled at days 9 (two of four guinea pigs) and 12 (three of four) with slight, focal, mostly portal hepatitis observed in two animals examined at day 9 and all animals at day 12 (minimal-to-mild). Signs of steatosis were largely absent in animals euthanized before day 9 post-infection. By day 12 post-infection, three of four animals displayed significant and diffuse steatosis ranging from mild to moderate-to-severe (Figure 8).

Analysis of lung specimens revealed the presence of minimal interstitial pneumonia in each animal across all time points in this study. However, the degree of inflammatory changes showed impressive progression. Minimal or mild-to-moderate changes were noted at days 1 and 3 post-infection, which progressed to severe or moderate-to severe changes in two of four animals examined at the 6- and 9-day time points and four of four animals by day 12. The relatively well-aerated alveoli were better preserved in guinea pigs examined at days 1 and 3, where their estimated average of the four animals was 62.5% and 60% in the examined samples, respectively. This value dropped to 51% of specimens examined from animals euthanized at 6 days and 45% at 9 days. The ratio of the open, well-aerated alveoli dramatically decreased further to an average of 11% in specimens examined from animals euthanized at 12 days post-infection at which point some animals scarcely had spacious, preserved alveoli (5%) without any compromise or damage by compression or intra-alveolar fluid (Figure 8). Pulmonary edema was found in all animals at the 12-day time-point though it was not present in any other groups.

Histological differences were less pronounced in the kidney and the spleen (Figure 8). The histology of the vascular and glomerular units and the renal medulla was unremarkable in all groups. Moderate mononuclear inflammation of the renal sinuses was recorded in animals euthanized on days 9 (subcalyceal in one, subcalyceal and perivenular in one animal) and 12 (subcalyceal in two animals) post-infection. The tubular injury was most significant animals from the 12-day group (mild in two animals, mild with occasional sloughed necrotic tubular epithelial cells in the other two) compared to the animals sampled at day 9 (mild in three animals, minimal in one). The other groups showed mixed findings including unremarkable renal tubules in animal euthanized on days 1 and 3 (one animal per time-point) and two animals sampled at day 6. The other animals in these groups displayed minimal or mild tubular injury. In spleen sections, significant necrosis was not detected across any of the groups. Heterophilic splenitis was displayed in one animal examined on day 6 (minimal), and all four animals euthanized on days 9 and 12 and ranged from minimal to mild. Focal lymphoid depletion was seen in one animal sampled at day 9 and in all four from day 12. Follicular hyperplasia or monomorphic infiltrate at the periphery of lymphoid follicles was observed in one animal at day 3 (minimal), all four animals at day 6 (mild), one animal at day 9 (minimal), and in all four animals at day 12 (mild-to-moderate).

## 4. Discussion

The high degree of genotypic and phenotypic variation that exists within LASV isolates from across West Africa highlights the importance of developing and evaluating disease models for LASV isolates from multiple countries [20,21]. Of particular importance is the development of convenient disease models for LF from countries like Nigeria, where annual outbreaks involving high case numbers are observed. Previously, our work has focused on using recent human isolates from Nigeria for the development of non-human primate and in-bred strain 13 guinea pig models [20]. Here, we sought to expand these studies towards the development of an outbred guinea pig model using commercially available Hartley guinea pigs, which can be utilized across several research facilities with high containment capabilities.

In our original studies, the NML-33 strain of LASV had a lethality rate of 83% in strain 13 guinea pigs with 5 of 6 animals infected succumbing to disease between 19- and 26-days post-infection [20]. Despite this, the NML-33 isolate did not result in uniform or consistent disease in pilot studies conducted in Hartley guinea pigs (data not shown). In contrast, preliminary studies found that a primary isolate from clinical materials (passage 1) for NML-61 achieved 50% lethality in Hartley guinea pigs. Within four additional passages, uniform lethality was observed. Similar to the findings with GPA-Josiah, the NML-61/GPA acquired only two dominant mutations, one within the NP and a second in the L proteins, during the adaptation process. Although this suggests the acquisition of lethality may be associated with increased replication, further experiments using a reverse genetics system is required to determine the precise genetic determinants of disease. Interestingly, despite the similar genetic differences, the LD_50_ for the NML-61/GPA (10 TCID_50_) was lower than that for GPA-Josiah (1000 TCID_50_) and disease progression with Hartley guinea pigs was more rapid [19].

Clinically, disease in animals infected with NML-61/GPA demonstrated the expected hallmarks of LASV infection including weight loss and increased body temperatures as well as high viral loads in tissue specimens examined [22,23,24]. Hematology revealed thrombocytopenia beginning 6 days post-infection. Consistent with this and supporting the onset of disseminated intravascular coagulopathy (DIC) were observations of increases in PT and aPTT which have been previously reported. Analysis of serum cytokine concentrations revealed an overall activation of pro-inflammatory mediators suggestive of an overactive host immune response to infection, particularly immediately preceding death. Analysis of serum biochemistries during the course of infection was largely unproductive. Increased GLOB levels were observed and may be due to the overactive immune response. Decreased levels of ALP and PHOS were also noted and may have been caused by malnutrition as suggested by overall weight loss. Decreased levels of ALB were the only observation consistent with liver disease and are also suggestive of increased vascular permeability associated with DIC. Notably, ALT levels remained static throughout the course of infection and the tests utilized in the current study did not include measurement of AST, a second key liver parameter.

Consistent with previous characterizations of guinea pig models of LF, the most striking histopathological findings were noted in the lungs and livers of infected animals, which was most pronounced immediately preceding death [21,22,23,24,25]. Between days 6 and 9 post-infection, lung and liver specimens from most animals demonstrated evidence of interstitial inflammation and steatosis. By day 12, the final time point before uniform lethality occurred, the extent of these abnormalities increased to moderate-to-severe and included pulmonary edema in lung specimens with little capability for oxygen exchange. Brain specimens were not evaluated as a part of the current study due to the lack of neurological signs of disease as well as the rapid time from infection to ensuing lethal outcome. However, subsequent studies evaluating this, as well as long-term sequalae including deafness could be considered using a sub-lethal inoculum.

Overall, the NML-61/GPA model described here provides a convenient small animal model for further study of a contemporary LASV isolate circulating in Nigeria. Although not all the hallmarks for human LF are noted in guinea pig models, notably indications of renal failure (increases in BUN and creatine), several markers are observed. Importantly, beyond weight and body condition assessment, several metrics are available including evaluation of body temperature, platelet levels, coagulation parameters, and host immune responses, that utilize non-lethal sampling with which one can monitor disease progression. Combined with terminal sampling analysis including viral loads and histopathology, this model is well positioned as a screening model to evaluate vaccine and medical countermeasures against clade III Nigerian LASV isolates.

## Figures and Tables

**Figure 1 viruses-15-00769-f001:**
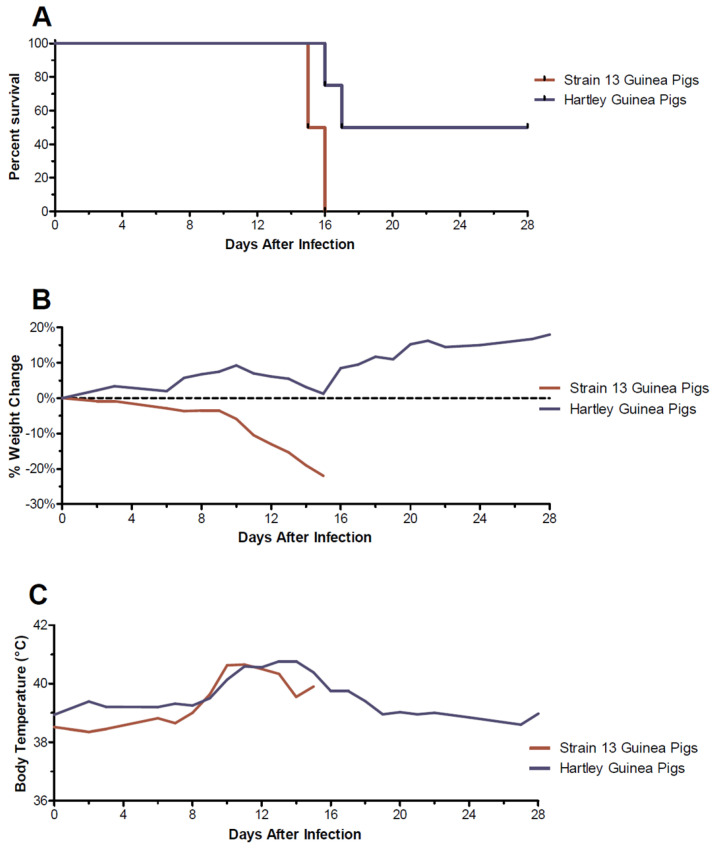
Comparison of clinical progression of a clade III Lassa virus isolate in strain 13 and Hartley guinea pigs. In an initial assessment of virulence, 6 strain 13 and 8 Hartley guinea pigs were inoculated with 1 × 10^5^ median tissue culture infectious doses of a p1 clinical LASV isolate and observed daily for signs of disease. Shown are percent survival curves (**A**), average percent weight change (**B**), and average body temperatures (**C**).

**Figure 2 viruses-15-00769-f002:**
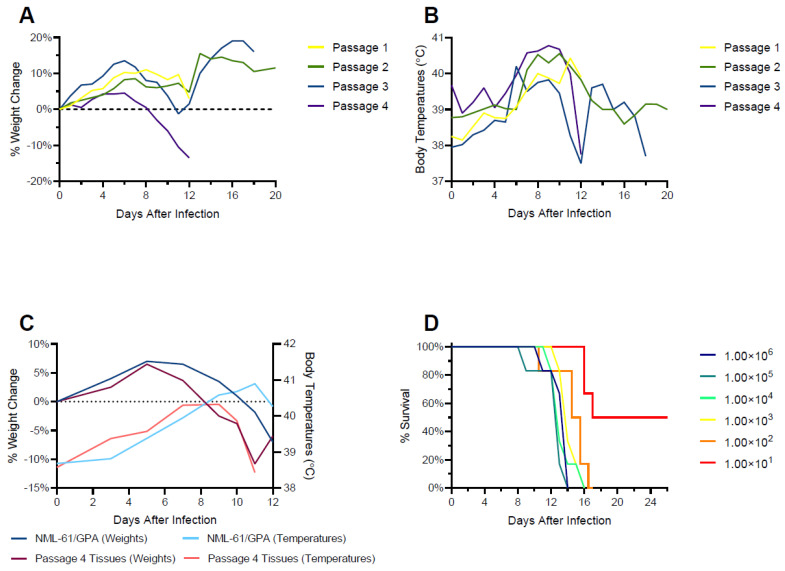
Guinea pig adaption of a clade III Nigerian Lassa virus isolate. Groups of 4 Hartley guinea pigs were infected with 1 × 10^5^ median tissue culture infectious doses of a primary isolate of a clade III Lassa virus isolate propagated on Vero E6 cells (passage 1) or 10% weight by volume pooled tissue homogenates collected from clinically ill or moribund animals from the previous passage (passages 2–4). Shown are percent weight change (**A**) and average body temperatures (**B**) observed post-inoculation per passage. (**C**) Comparison of weight change and body temperatures from 6 guinea pigs inoculated with passage 4 tissues or a cell culture isolate (NML-61/GPA) derived from passage 4 tissues. (**D**) Determination of the median lethal dose of guinea pigs inoculated with a cell culture derived, guinea pig adapted clade III Lassa virus.

**Figure 3 viruses-15-00769-f003:**
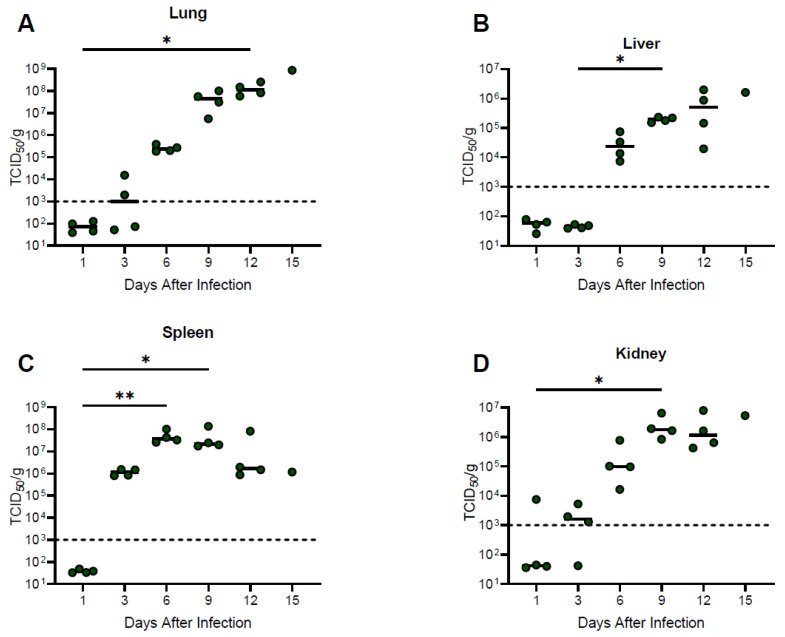
Temporal analysis of infectious viral loads from guinea pigs infected with NML-61/GPA. Twenty-four guinea pigs were inoculated with 100LD_50_ of NML-61/GPA. Groups of four animals were euthanized at indicated days post-inoculation and Lung (**A**) Liver (**B**) Kidney (**C**) and Spleen (**D**) samples collected and analyzed by standard cell culture techniques for the presence of infectious Lassa virus. Dots represent individual animal values, bar represents group averages. Dashed line represents the lower limit of confidence in the infectious assay. Significant differences where *p* < 0.05 and <0.01 are indicated by * and **, respectively.

**Figure 4 viruses-15-00769-f004:**
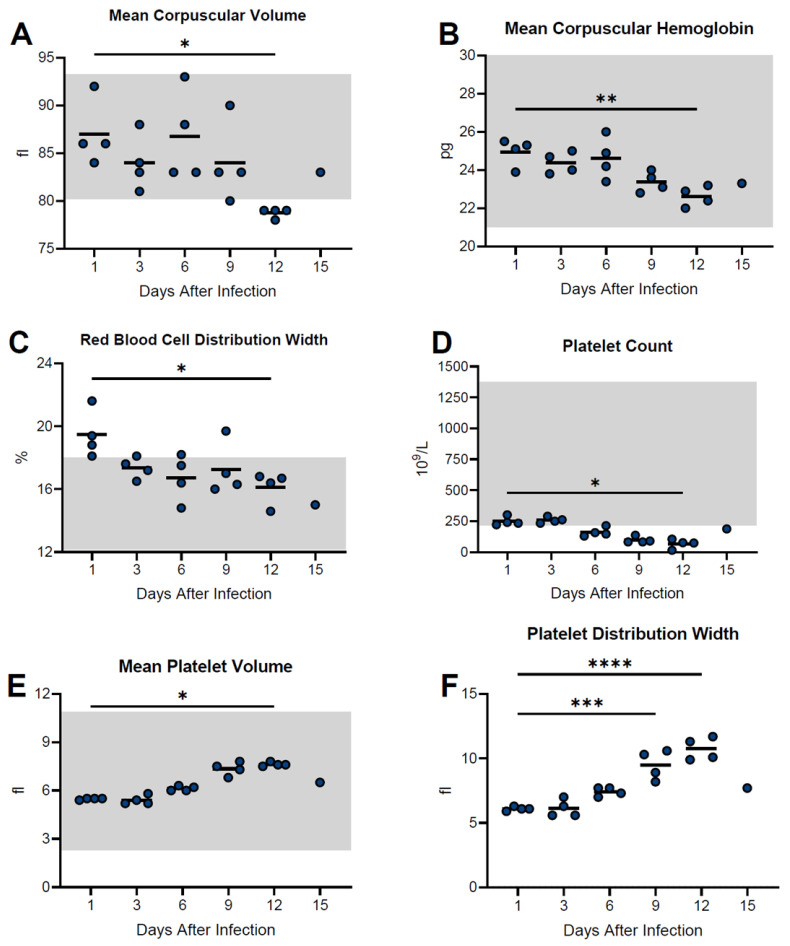
Temporal analysis of hematological values from guinea pigs infected with NML-61/GPA. Twenty-four guinea pigs were inoculated with 100LD_50_ of NML-61/GPA. Groups of four animals were euthanized at indicated days post-inoculation and analyzed for hematological abnormalities. Shown are data for mean corpuscular volume (**A**), mean corpuscular hemoglobin (**B**), red blood cell distribution width (**C**), platelet count (**D**), mean platelet volume (**E**), and platelet distribution width (**F**). Dots represent individual animal values; bar represents group averages. Where available, normal ranges are indicated by gray shading (www.criver.com, accessed 8 March 2023). Significant differences where *p* < 0.05, <0.01, <0.001, and <0.0001 are indicated by *, **, ***, and **** respectively.

**Figure 5 viruses-15-00769-f005:**
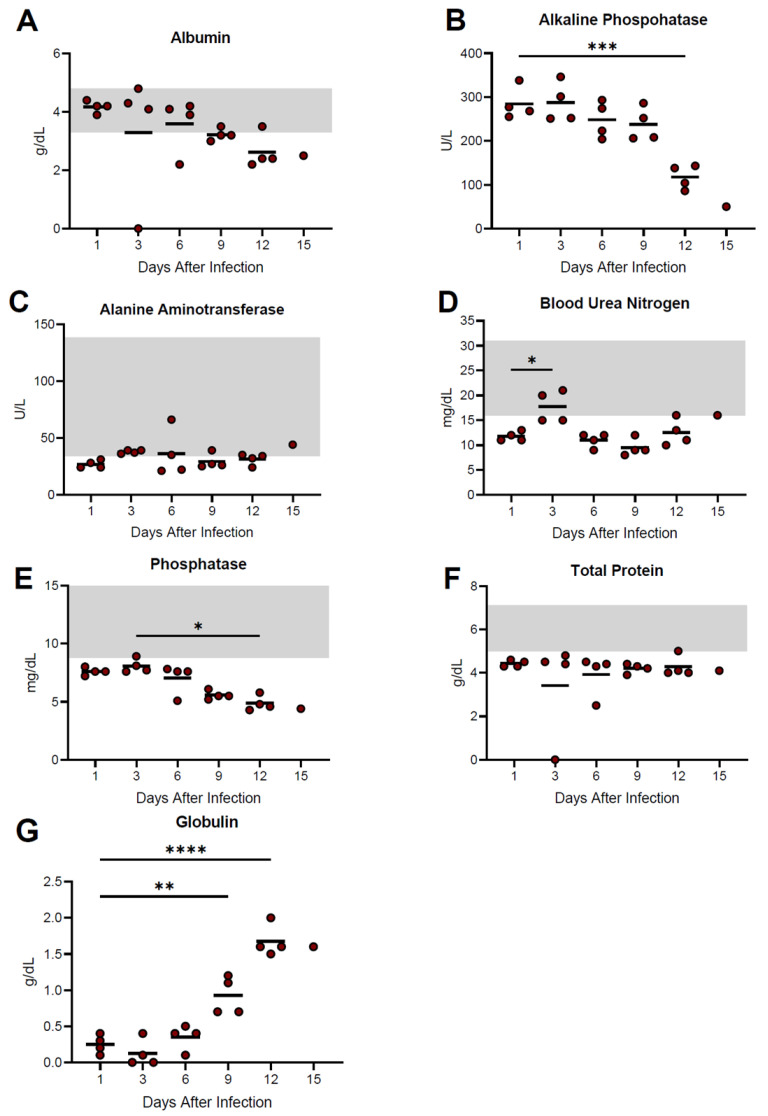
Temporal analysis of serum biochemistries from guinea pigs infected with NML-61/GPA. Twenty-four guinea pigs were inoculated with 100LD_50_ of NML-61/GPA. Groups of four animals were euthanized at indicated days post-inoculation and analyzed for biochemical abnormalities. Shown are data for albumin (**A**), alkaline phosphatase (**B**), alanine aminotransferase (**C**), blood urea nitrogen (**D**), phosphataI (**E**), total protein (**F**), and globulin (**G**). Dots represent individual animal values, bar represents group averages. Where available, normal ranges are indicated by gray shading (www.criver.com, accessed 8 March 2023). Significant differences where *p* < 0.05, <0.01, <0.001, and <0.0001 are indicated by *, **, ***, and **** respectively.

**Figure 6 viruses-15-00769-f006:**
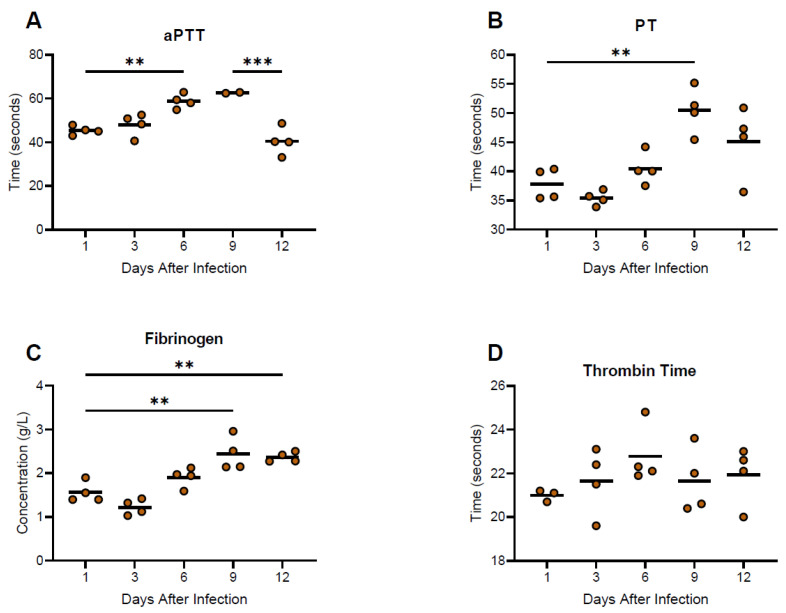
Coagulation profiles from guinea pigs infected with NML-61/GPA. Twenty-four guinea pigs were inoculated with 100LD_50_ of NML-61/GPA. Groups of four animals were euthanized at indicated days post-inoculation and analyzed for aPPT (**A**), PT (**B**), Fibrinogen (**C**), and Thrombin time (**D**). Dots represent individual animal values; bar represents group averages. Significant differences where *p* < 0.01, and <0.001, are indicated by **, and ***, respectively.

**Figure 7 viruses-15-00769-f007:**
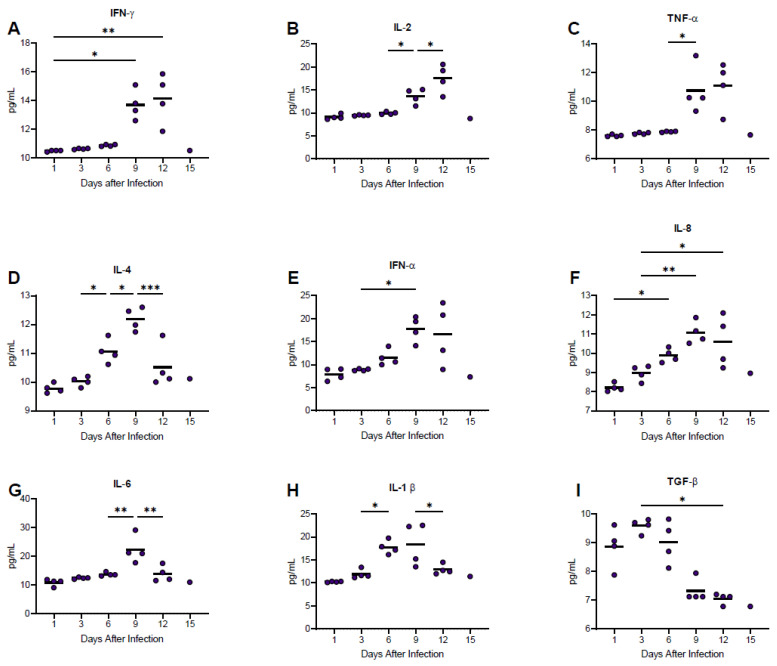
Temporal analysis of host cytokine responses in guinea pigs infected with NML-61/GPA. Twenty-four guinea pigs were inoculated with 100LD_50_ of NML-61/GPA. Groups of four animals were euthanized at indicated days post-inoculation and analyzed for IFN-gamma (**A**), IL-2 (**B**), TNF-alpha (**C**), IL-4 (**D**), INF-aIha (**E**), IL-8 (**F**), IL-6 (**G**), IL-1 beta (**H**), and TGF-beta (**I**). Dots represent individual animal values; bar represents group averages. Significant differences where *p* < 0.05, <0.01, and <0.001 are indicated by *, **, and ***, respectively.

**Figure 8 viruses-15-00769-f008:**
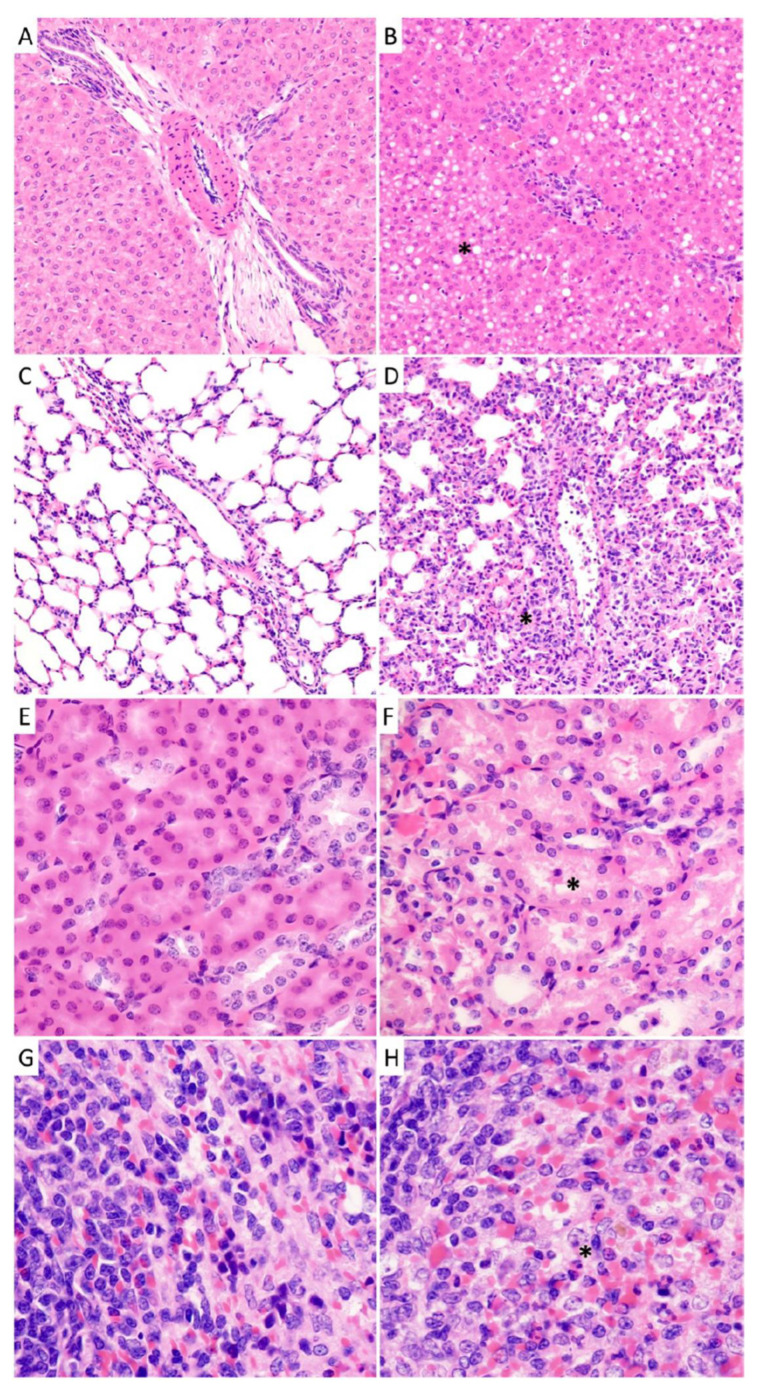
Histopathological analysis of tissues collected guinea pigs infected with NML-61/GPA. Guinea pigs were inoculated with 100LD_50_ of NML-61/GPA and euthanized at regular intervals for histological analysis of solid organs. Shown are representative liver (**A**,**B**), lung (**C**,**D**), kidney (**E**,**F**), and spleen (**G**,**H**) specimens collected early (day 1, (**A**,**C**,**E**,**G**)) post-infection as well as when animals were moribund (day 12, (**B**,**D**,**F**,**H**)). Primary observations included significant and diffuse steatosis in liver sections (**B**) as well as any compromised and damaged alveoli with intra-alveolar fluid in lung sections (**D**) collected from moribund animals.

**Table 1 viruses-15-00769-t001:** Summary of mutations.

Coding Region	Position	Mutation	Frequency	Coding Region	Position	Mutation	Frequency
L	1221	Asn→Asp	58.7%	GPC	213	Met→Ile	11%
L	672	Lys→Glu	14.7%	NP	435	Val→Ile	6.1%
L	607	Gly→Glu	13.8%	NP	228	Ile→Val	53.5%
L	515	Thr→fs	6%	NP	62	Gly→Asp	14.9%

## Data Availability

Data is contained within the article or Appendix A.

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
