# Peer review of "An Outbred Guinea Pig Disease Model for Lassa Fever Using a Host-Adapted Clade III Nigerian Lassa Virus"

_viruses, 2023, doi:10.3390/v15030769_

Round 1

Reviewer 1 Report

Deschambault and co-authors have adapted a Clade III Lassa virus into the Hartley guinea pig. The study is well designed. The development of this model allows enables future studies looking at prophylactics and disease modeling. And an important step for further understanding Lassa fever.

I only have a few small comments I would like to see addressed.

Line 46-47, Ribavirin is the only approved treatment but a few are in clinical trials. I would like to see a comment on those.

Lines 167-177, Is it known whether the virus used in these studies was from a fatal case of Lassa fever?

 In the discussion-

The authors do well to compare the similarities in disease cause by the new clade III vs. the Josiah strain but they do not address similarities or difference between the viruses themselves. How does this clade III virus compare to the Guinea pig adapted Josiah virus? In regards to the mutations generated through adaptation to the Hartley model. 

Author Response

Line 46-47, Ribavirin is the only approved treatment but a few are in clinical trials. I would like to see a comment on those.

Response: We acknowledge that there are modalities currently in clinical trials for Lassa fever; however further discussion on these would not strengthen this paper. Merely mentioning them does not change the statements that there are currently no treatments approved and models like the one described here are required for initial evaluation of potential MCMs.  

Lines 167-177, Is it known whether the virus used in these studies was from a fatal case of Lassa fever?

Response: Yes, this is indicated in the reference, and we have added “non-lethal” to the statement describing the isolate for clarity. 

 In the discussion-

The authors do well to compare the similarities in disease cause by the new clade III vs. the Josiah strain but they do not address similarities or difference between the viruses themselves. How does this clade III virus compare to the Guinea pig adapted Josiah virus? In regards to the mutations generated through adaptation to the Hartley model. 

Response: In the discussion we mention the similarity in mutations between the GPA-Josiah and our NML-61/GPA virus. Both adapted viruses have dominant mutations within the NP and L suggesting the acquisition of lethality may reside in the replication complex. Before delving too deeply into this supposition though we feel it would be necessary to introduce these mutations into a reverse genetics derived virus to determine the molecular basis of lethal disease. We have added some comments on this in the discussion section of the manuscript.  

Reviewer 2 Report

Deschambault and colleagues describe a new small animal model for Lassa fever based on a lineage III strain from Nigeria. The lethality rate of the strain was increased by serial passage in outbred guinea pigs. During passage, multiple mutations were selected in smaller populations of the virus. The model will be useful for preclinical testing of Lassa fever vaccines and drugs. The results are well described, presented in the figures, and discussed. I have only a few minor comments:

1. Some of the characteristic features of Lassa fever in humans are not reproduced in the model, including increases in liver enzymes (although AST was not tested) and renal failure (increases in BUN and creatinine). The authors should briefly address this limitation.

2. The virus phenotype appears to be associated with several genotypic changes, most of which are found only in small variant populations. It is not clear whether all or some of these mutations are linked on the same genome or whether the virus population consists of a mixture of different minor variants. It is therefore not possible for others to recreate the virus population that produces the observed phenotype. This lack of knowledge limits the reproducibility of the model. This limitation should be mentioned in the discussion. It is recommended that a future study clone and characterize the variants that make up the virus population and determine which variant(s) are associated with the observed lethal phenotype.

Author Response

Response: We thank the reviewer for the positive comments.

  1. Some of the characteristic features of Lassa fever in humans are not reproduced in the model, including increases in liver enzymes (although AST was not tested) and renal failure (increases in BUN and creatinine). The authors should briefly address this limitation.

Response: We have addressed this point with the addition of the following in the discussion “Overall, the NML-61/GPA model described here provides a convenient small animal model for further study of a contemporary LASV isolate circulating in Nigeria. Although not all the hallmarks for human LF are noted in guinea pig models, notably indications of renal failure (increases in BUN and creatine), several markers are observed. Importantly, beyond weight and body condition assessment, several metrics are available including evaluation of body temperature, platelet levels, coagulation parameters, and host immune responses, that utilize non-lethal sampling with which one can monitor disease progression.”

  1. The virus phenotype appears to be associated with several genotypic changes, most of which are found only in small variant populations. It is not clear whether all or some of these mutations are linked on the same genome or whether the virus population consists of a mixture of different minor variants. It is therefore not possible for others to recreate the virus population that produces the observed phenotype. This lack of knowledge limits the reproducibility of the model. This limitation should be mentioned in the discussion. It is recommended that a future study clone and characterize the variants that make up the virus population and determine which variant(s) are associated with the observed lethal phenotype.

Response: We agree with the reviewer that the impact of the several minor and few dominant mutations observed are unclear at this time and will need further in-depth analysis using reverse genetics to better elucidate them. However, we disagree that this limits the reproducibility of the model. Most in vivo disease modelling efforts do not use plaque purified, genetically homogenous, viral inoculums, but rather a heterogenous, quasispecies, challenge preparation that contains minor and dominant genetic differences that are hidden behind the often-reported consensus sequences. Nevertheless, we intend to further delve into the genetic determinants of the lethal phenotype observed in this model as well as the guinea pig adapted Lassa virus Josiah model as the reviewer has suggested and have added some discussion on that to the paper.   

Reviewer 3 Report

In this study, authors have inoculated guinea pigs with NML-61, a lineage II Lassa strain isolated from a symptomatic case in Nigeria in 2018. They used strain 13 and Hartley guinea pigs. Strain 13 guinea pigs are known to be susceptible to many virus while Hartley are usually more resistant. Nevertheless, strain 13 guinea pigs are hardly available while Hartley guinea pigs are commercially available. As expected, NML-61 is uniformly lethal in strain 13 guinea pigs while it causes 50% in Hartley guinea pigs. Authors have then adapted NML-61 to guinea pigs by serial passaging and obtained a guinea pig adapted NML-61/GPA strain causing uniform lethality in Hartleys. NML-61/GPA Hartleys develop a disease that resembles the disease observed in guinea pigs using other Lassa strains, including clinical signs such as fever and weight loss, increased virus replication in several organs, coagulation disorders and modification of the blood biochemistry. Therefore, authors here provide a new model of Lassa virus infections that will be useful for the evaluation of vaccines or other medical countermeasures.

The manuscript is well written and the figures of quality. The subject is of interest as there is an increased need for small animal models due to the scarcity of non human primates in the recent context. These small animal models can indeed be very useful for vaccine and antiviral testing. Nevertheless, I have several comments that would need to be addressed before recommending publication.

Major comments:

1.      Authors should provide results for control uninfected animals. Without this, it is difficult to interpret the impact of virus replication on the biochemical, histological and coagulation parameters. Historical controls may be used.

2.      Authors should provide a measure of viremia as increased viremia is a hallmark of Lassa virus infections.

Minor comments:

1.      Lines 222, 223, 224: authors should qualify heir statements on temperature as for the tissue-grown NML-61/GPA, the body temperature does not even reach 39°C.

2.      Line 264: a significant decrease in the albumin concentration is noted at day 12 but there is no statistics on the albumin graph.

3.      Figure 7: correct INF-α for IFN-α.

4.      Lines 340-357: a long description of the histopathological findings in the kidney and spleen is presented without any supporting figure. Authors should provide figures (supplementary materials?) otherwise this description is anecdotal.

Author Response

Major comments:

  1. Authors should provide results for control uninfected animals. Without this, it is difficult to interpret the impact of virus replication on the biochemical, histological and coagulation parameters. Historical controls may be used.

Response: Where available, we have added normal ranges for specific markers as determined from healthy guinea pigs to the figures.

  1. Authors should provide a measure of viremia as increased viremia is a hallmark of Lassa virus infections.

 Response: Due to volumes of serum required for the hematological, biochemical and coagulation analysis, we were unable to perform infectious titrations on blood or serum samples. However, during the course of the study, viremia was monitored in real-time using a qualitative molecular test. This method and the below result have been included in the revised manuscript.

“Consistent with these observations, RT-PCR positive blood samples were first noted on day 3 post-infection (3 of 4 animals), with uniform positivity detected on and after day 6 post-infection. Although the molecular assay was not quantitative, the decreasing Ct values noted after day 6 imply increasing viral titers up until the terminal time point.” 

Minor comments:

  1. Lines 222, 223, 224: authors should qualify heir statements on temperature as for the tissue-grown NML-61/GPA, the body temperature does not even reach 39°C.

Response: We thanks the reviewer for pointing this out. The figure itself is the issue and not the text. The figure has been revised and is indeed correct now.  

  1. Line 264: a significant decrease in the albumin concentration is noted at day 12 but there is no statistics on the albumin graph.

Response: The reviewer is correct and the mention of a significant decrease in ALB at day 12 has been removed from the text.

  1. Figure 7: correct INF-α for IFN-α.

Response: Corrected. Thank you for pointing this out.

  1. Lines 340-357: a long description of the histopathological findings in the kidney and spleen is presented without any supporting figure. Authors should provide figures (supplementary materials?) otherwise this description is anecdotal.

Response: We have added images of spleen and kidney to the histopathology figure as requested.

Reviewer 4 Report

This is really impressive work. Three concerns need to be clarified before acceptance: 1. Is it possible to get an adapted virus with a shorter death time?  2. The adapted NML-61/GPA virus is a hybrid virus, containing many viruses carrying different mutations. Why did the isolation of a single virus not continue? In addition, is this virus stable in successive passages, and is there a possibility of revertant mutations? 3. Are these two dominant mutations the determinants of enhanced virulence?

Author Response

Responses: We thank the reviewer for the comments which hare sequentially addressed below.

  1. Shorter time to death. In theory it may be possible to adapt a virus to have a reduced time to lethal disease, however it is unclear what this would accomplish. This model seems to cause lethal disease in a similar timeframe to other guinea pig models for LF. The prolonged disease progression observed in these animals over that of mice provides a larger and more realistic therapeutic window for which to evaluate medical countermeasures. We have not added any discussion on this to the revised paper.
  2. Plaque purify. I think the reviewer is suggesting that we should plaque purify the adapted virus and proceed with a single homogenous viral inoculum. For the purpose of model development this is not a necessary step however, we agree that is necessary to define the genetic determinants of disease in guinea pigs and have plans to do such studies. We have added some discussion on this to the revised paper.
  3. Reverse genetics. In short, we do not know the answer to this question yet. In order to determine this, we require the use of a reverse genetics system, which we now have in place. Future studies will aim to better define the genetic determinants of disease associated with the mutations observed in the current study as well as previous work with a guinea-pig adapted Josiah virus.

Round 2

Reviewer 3 Report

Authors have addressed all the comments in the text but figure modifications do not appear in the revised version. In addition, technical details should be provided on the RT-PCR assay in the methods section (primers,...). Authors mention a qualitative RT-PCR but describe CT values suggesting RT-qPCR. Please clarify. Results of the PCR assay should also be provided as supplementary material.

Author Response

As requested, we have added the PCR data as a supplemental table (see final page of manuscript). The reviewer is correct that the test is a real-time assay but it is a screening assay and not quantitative. Unfortunately, the samples have been stored since prior to the covid-19 pandemic (when the animal work was completed) and so are not of sufficient quality to re-run on a quantitative assay.  

The revised figures are with the editorial staff and should be embedded into the revised paper shortly. We were unable to do it on our end as it distorted the files. 
